# Frequent and Automatic Update of Lane-Level HD Maps with a Large Amount of Crowdsourced Data Acquired from Buses and Taxis in Seoul

**DOI:** 10.3390/s23010438

**Published:** 2022-12-31

**Authors:** Minwoo Cho, Kitae Kim, Soohyun Cho, Seung-Mo Cho, Woojin Chung

**Affiliations:** 1Department of Mechanical Engineering, Korea University, Seoul 02841, Republic of Korea; 2School of Mechanical Engineering, Purdue University, West Lafayette, IN 47907, USA; 3SK Telecom Co., Ltd., Seoul 04539, Republic of Korea

**Keywords:** HD map, change detection, mapping, autonomous vehicle navigation, segmentation and categorization, clustering, crowdsourcing

## Abstract

Recently, HD maps have become important parts of autonomous driving, from localization to perception and path planning. For the practical application of HD maps, it is significant to regularly update environmental changes in HD maps. Conventional approaches require expensive mobile mapping systems and considerable manual work by experts, making it difficult to achieve frequent map updates. In this paper, we show how frequent and automatic updates of lane marking in HD maps are made possible with enormous crowdsourced data. Crowdsourced data is acquired from onboard low-cost sensing devices installed on many city buses and taxis in Seoul, South Korea. A large amount of crowdsourced data is daily accumulated on the server. The quality of sensor measurement is not very high due to the limited performance of low-cost devices. Therefore, the technical challenge is to overcome the uncertainty of the crowdsourced data. Appropriately filtering out a large amount of low-quality data is a significant problem. The proposed HD map update strategy comprises several processing steps including pose correction, observation assignment, observation clustering, and landmark classification. The proposed HD map update strategy is experimentally verified using crowdsourced data. If the changed environments are successfully extracted, then precisely updated HD maps are generated.

## 1. Introduction

Nowadays, high definition (HD) maps are used in various engineering applications. HD maps can accurately represent multiple kinds of road information, such as traffic lights, traffic signs, lane markings, and road structures. In city planning, this information is used for infrastructure design and maintenance [1,2]. HD maps also play significant roles in the construction of geospatial data in augmented reality (AR) [3]. In particular, detailed HD maps can significantly improve the safety and accuracy of autonomous driving [4,5]. Above all else, HD maps have become important elements of autonomous vehicles, from localization [6,7] to object detection [8] and path planning [9,10].

One of the major challenges with utilizing HD maps is its costly maintenance. Because environmental changes frequently occur in urban areas, the demand for keeping HD maps up-to-date is increasing in practical applications. For safe navigation and robust localization of autonomous cars, timely HD map updates are required. Conventional approaches require high-cost mobile mapping systems [11] and a lot of manual work by experts [12]. It is noted that, although diverse heuristic-based [13,14] and machine learning-based [15,16,17,18] procedures have been introduced, it is still difficult to find a practical solution for the HD map update problem.

Recently, many companies have considered crowdsourced data as a starting point for performing HD map updates [19,20,21,22]. Distributed vehicles equipped with crowdsourcing devices could travel numerous road environments in a short time at a relatively low cost. Therefore, crowdsourced data is typically considered to have great potential as a source of road information for keeping HD maps up-to-date. However, previous studies with crowdsourced data have mainly focused on affordable HD map generation [23,24,25,26,27,28,29,30,31,32]. In addition, some studies have addressed HD map updates by using high-cost sensors [33,34,35,36], which are not directly applicable to a large-scale crowdsourcing scenario. As for HD map updating, relatively few studies have been conducted by leveraging crowdsourced data.

For HD map updates, landmark observation and global localization are essential. Therefore, crowdsourcing devices typically involve a low-cost camera and global navigation satellite system (GNSS) receiver. In order to utilize crowdsourced data successfully, it is required to focus on its high uncertainty. The quality of sensor measurement is not very high due to the limited performance of low-cost devices. Additionally, many hard technical problems exist, such as occlusions by dynamic obstacles and blurred observations caused by vehicle movement.

Several methods have been proposed to update HD maps using crowdsourced data. Sun et al. suggested a geometric HD map change detection method in highway roads [37]. They surveyed a lane width between the curb and reference lane to detect road changes, which showed robustness against low-precision measurements. Pannen et al. detected HD map changes in motorway construction sites using a boosted particle filter [38]. Their method suggested two different particle filter setups to grant robustness against environmental interruptions. Pannen et al. also proposed an HD map update framework, which managed crowdsourced data using a standard definition (SD) map [39]. They partitioned the data according to basic road structures using the SD map, and then traffic lanes were updated by filtering candidates observed at less than a certain threshold. Their method requires frequent updates of SD maps, which is not easy to obtain in the real world. Massow et al. proposed a technical architecture, which updated traffic lanes in highway sections [40]. Kim et al. [41] and Dubois et al. [42] accurately updated traffic signs on simulation scenarios.

To the best of the author’s knowledge, few studies have leveraged an enormous amount of crowdsourced data to maintain the latest HD maps. In addition, most existing methods have been tested in highway or simulation scenarios in which environmental interruptions are relatively less than in urban areas. In urban canyons, various kinds of road segments exist, including straight sections, curves, intersections, construction sites, crosswalks, and bus boarding zones, which makes HD map updates more challenging. Furthermore, crowdsourced data might be more subject to environmental disturbances due to a large volume of dynamic obstacles and blockages of GNSS signals by high buildings. When considering the deployment of crowdsourcing devices in urban areas, overcoming various uncertainty of crowdsourced data becomes way more important.

In this paper, it is shown how frequent and automatic updates of HD maps are made possible with a large amount of crowdsourced data. The main interest of our research is to update lane-level landmarks. This is because changes in lane segments can directly affect the driving route for autonomous driving [43]. We suggest practical strategies in order to deal with an enormous amount of crowdsourced data without reference to precise prior, as well as for further application to complex urban areas. As a sophisticated extension of the previous work [44], a variety of processing steps are integrated, including pose correction, validation check, observation assignment, observation clustering, and landmark classification. Specifically, we improve low-quality data processing with a sequence of filtering methods and recursive Bayesian estimation. This procedure should be carefully implemented considering data characteristics. If the changed environments are successfully extracted, then precisely updated HD maps are generated.

The usefulness of the proposed scheme is experimentally verified with a large amount of crowdsourced data acquired from many city buses and taxis in Seoul. We utilize crowdsourced data, which is called road observation data (ROD). The Korean IT company, SK Telecom, has collected ROD in Seoul using 1600 public buses and 100 taxis since 2020 [22]. In fact, a large amount of crowdsourced data can be acquired from onboard sensing devices, and then accumulated on the server every day.

## 2. Crowdsourced Data

In this section, we explain the characteristics of the HD map and crowdsourced data utilized in this paper. We employed an HD map that is open to the public by the National Geographic Information Institute of Korea. The HD map includes a visual landmark layer, which contains traffic lights, traffic signs, lanes, and road connectivity. Our HD map also consists of nodes and edges. Because we focus on urban environments, each node corresponds to a road segment or intersection, as in [44].

Road observations have been collected from crowdsourcing devices, consisting of a low-cost camera and a GNSS sensor. Figure 1 shows the road observations of a single drive. Keyframes represent the discrete 3D poses of a crowdsourcing device during a single drive sequentially. Observations include observed continuous and discrete landmarks. Observations are referenced to the local keyframe poses. If the keyframe poses and observations are sufficiently accurate, then Figure 1 accurately describes the environmental geometry. Each landmark point represents the degree of uncertainty as a covariance ellipse.

The total travel distance and driving time of all the crowdsourcing vehicles are approximately 89,620 km and 3600 h every day. Furthermore, the total number of lane observations is approximately 8 billion points. Figure 2 shows the quantitative information of the ROD collected from five test areas that were randomly sampled in Seoul. In this paper, the ROD collected on 5 July 2021, is employed. For the case of Test area 1 in Figure 2, the road length is 1.6 km, and the total travel distance is 503.59 km. This result implies that many vehicles traversed the area more than 300 times per day. We can collect plenty of observations for broad urban environments.

## 3. Frequent and Automatic Update of HD Maps

### 3.1. Overview of Map Update

An overview of the proposed map update scheme is shown in Figure 3. The first step is to load the ROD and HD map for the area of interest. Second, the estimated poses of the ROD are corrected and reliable observations are selected through validation checks. In the third step, observations are assigned to the visible landmarks of the HD map. Once observations are assigned, the non-assigned observations are clustered with each other. Finally, based on the results of the observation learner, landmark classification is performed to detect changes in the given HD map. When only the changed environments are clearly extracted, the landmark states are updated in the HD map.

### 3.2. Loading ROD and HD Map

To effectively utilize a large amount of data on map updates, ROD is divided into section *i* and by vehicle *j*. The ROD is originally divided into tiles with a width of approximately 2.44 km, with respect to latitude and longitude. We further partition the ROD by each section that corresponds to a 50–60 m long road segment or intersection. Additionally, the number of data collecting vehicles is about 1700. Most vehicles collect ROD along the bus routes in Seoul.

$M_{l}$ and $O_{l}$ denote the local HD map and ROD in Equation (Equation 1). $M_{l}$ is composed of local sub-maps $m_{i}$, and $O_{l}$ includes observations $o_{ij}$.
(1)$$\begin{array}{c} M_{l}=\left\{m_{i}\right\},O_{l}=\left\{o_{ij}\right\} \end{array}$$i: section  index,j: vehicle index


Thus, when the local HD map and ROD are loaded, sub-maps and observations are loaded. The components of $M_{l}$ and $O_{l}$ can differ according to the region of interest.

### 3.3. Observation Correction

#### 3.3.1. Pose Correction

The pose correction step is required to overcome the positional uncertainty of crowdsourced data. Keyframes are essential for map updates because observations are referenced to the pose of the keyframes. When the keyframe poses are inaccurate, the subsequent procedures result in low-quality updated maps.

Keyframe poses are estimated in real time through a sequence of internal processing steps in crowdsourcing devices. This process includes dead reckoning based on IMU and the wheel speed of the vehicles. In addition, semantic localization further increases the degree of accuracy. After all the localization procedures are completed, crowdsourced data is stored on a server. However, since pose estimation is conducted online using a low-cost device, an offline pose correction process can greatly contribute to the improvement in localization accuracy.

To correct the position in the background, a smoothing procedure is conducted based on the relative dead reckoning and keyframe poses of ROD. The g2o library [45] is utilized for optimization. The next step is to correct the lateral position through lane matching.

#### 3.3.2. Validation Check

In the validation check step, the ROD is evaluated to select reliable observations. When the lane-matching errors between the observations and HD map are higher than a predefined threshold, the observations are discarded. The reference value was designed by considering the average gap between lanes.

### 3.4. Observation Learner

#### 3.4.1. Observation Assignment

Even after the observation correction step, numerous uncertain observations may remain because of the large amount of data gathered in urban environments. Hence, an additional strategy for filtering out observations is necessary for accurate map updates.

In our crowdsourced data, the uncertainty of the observations becomes high when a lane is far away from the keyframes, as shown in Figure 1. Therefore, the range of the visible area referenced to the keyframe determines the quality and number of observations exploited for map updates. For example, the larger the visible area, the more observations are considered, but many uncertain observations are also included. Otherwise, fewer high-quality observations could be used.

The quality and coverage of updated maps are affected by the determination of visible landmarks. In our previous work [44], the observation assignment step mainly focused on the assignment of observations to HD map landmarks that showed the highest correspondence. However, we augment our algorithm with discreet consideration of the visible landmarks.

Algorithm 1 shows the overall procedure for searching visible landmarks, as described in Figure 4. In Figure 4a, we denote the reference and comparison pointsets as $P_{ref}$ and $P_{com}$, where *i* and *j* correspond to the lane index and point index of the $i_{th}$ lane, respectively.
(2)$$\begin{array}{c} P_{ref}=\left\{a_{i,j}\right\},P_{com}=\left\{b_{i,j}\right\} \end{array}$$$$i:\;\mathrm{lane}\;\mathrm{index},j:\;\mathrm{point index of the}\;i_{\mathrm{th}}\;\mathrm{lane}$$


In addition, the keyframe of $P_{com}$ is denoted by $p_{key}$, which consists of $k_{o}$ and ${\hat{k}}_{d}$. $k_{o}$ represents the position and ${\hat{k}}_{d}$ corresponds to the direction vector. In the assignment step, $P_{ref}$ and $P_{com}$ correspond to the lanes of the HD map and observations from the ROD, respectively.
**Algorithm 1:** Searching procedure for visible landmarks.
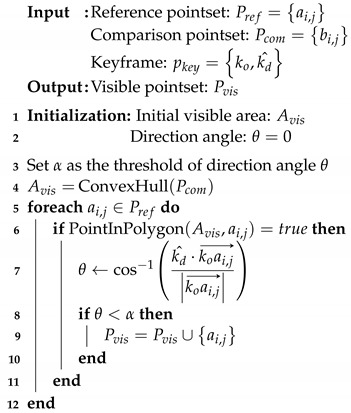


   First, the initial visible area $A_{vis}$ is constructed using the convex hull [46] of a set of points in $P_{com}$, as shown in line 4 of Algorithm 1 and Figure 4b. As a next step, if any point $a_{i,j}$ of $P_{ref}$ lies inside the boundary of $A_{vis}$, the direction angle θ is computed for $a_{i,j}$, as shown in line 7 of Algorithm 1 and Figure 4c. In this process, the dot product of vectors ${\hat{k}}_{d}$ and $\overset{⟶}{k_{o}a_{i,j}}$ is used. Consequently, when θ is less than the threshold α, $a_{i,j}$ is added to $P_{vis}$. The number of points in $P_{vis}$ is adjusted using the value of α. Thus, $P_{vis}$ is determined as presented in Figure 4d.

The proposed searching algorithm is advantageous for precisely finding visible landmarks, albeit conservatively. Because the distribution of observations for each keyframe is irregular owing to the use of low-cost sensors, the searching process for the visible area is essential. This process is available because of the large number of observations from crowdsourced data.

When the search process is completed, the comparison lanes of $P_{com}$ are assigned to the visible reference lanes of $P_{vis}$, using the lane-learner algorithm in [44]. The shell structure for graph fitting in [47] was modified to assign and cluster the landmarks. According to the shell structure shown in Figure 5, this lane-learner algorithm decides whether the comparison lane is assigned to the reference lane. When the average Euclidian distance between matched points is below a certain threshold, these two lanes belong to the same continuous landmark.

One of the advantages of the lane-learner algorithm is that the assignment performance is consistent regardless of the lane shape. The comparison lanes are successfully assigned to diverse lanes, including straight, curved, and split lanes. Furthermore, this lane-learner algorithm showed superior performance in the assignment of closely-parallel lanes compared with DBSCAN [48], a well-known lane point clustering method.

#### 3.4.2. Observation Clustering

In the observation clustering step, observations that are not assigned with HD map landmarks are handled. The clustering method is the same as that of the lane-learner algorithm described in Section 3.4.1. In this section, both the reference and comparison lanes are selected from unassigned landmarks.

If unassigned landmarks are repeatedly clustered, it is highly likely that newly added landmarks will exist in the HD map. To reduce the difference with the ground truths, candidates are recursively clustered until an overlap between the candidates does not occur.

### 3.5. Landmark Update

#### 3.5.1. Landmark Class Classification

To reflect changes in the HD map, landmark classes were defined as [44]. Table 1 lists the four landmark classes according to their existence, before and after the map update.

*Normal* and *Deleted* are the classes existent on the HD map before the update. After the update, *Normal* is maintained, whereas *Deleted* is removed from the map. In the case of *New* and *Outlier*, they correspond to clustered observations from the crowdsourced data. When the update process is completed, *New* is newly created on the map.

Classification procedures are different according to the type of landmarks. *Normal* and *Deleted* are decided by means of assignment results from Section 3.4.1. In [44], we used the fraction between the number of successful assignments and crowdsourced data. However, this simple criterion is not sufficient in urban environments because of the large amount of uncertain crowdsourced data. Thus, in this paper, we further adopt a recursive Bayesian model to estimate the state of landmarks in a more accurate and sophisticated manner.

Algorithm 2 shows the state estimation process for an existing landmark on the HD map. The Bayesian model is defined as follows:(3)$$\begin{array}{c} \mathrm{bel}\left(x_{t}\right)=p\left(x_{t}|z_{1:t}\right)=\eta \;p\left(z_{t}|x_{t}\right)\mathrm{bel}\left(x_{t-1}\right)\\ where\;\;\;\eta =1/p\left(z_{t}|z_{1:t-1}\right)\\ x_{t}\in \left\{1,0\right\}\;\left(1:\mathrm{exist},0:\mathrm{do}\;\mathrm{not}\;\mathrm{exist}\right)\\ z_{t}\in \left\{1,0\right\}\;\left(1:\mathrm{detect},0:\mathrm{do}\;\mathrm{not}\;\mathrm{detect}\right) \end{array}$$

Equation (Equation 3) defines the posterior belief of the landmark’s existence at time *t*, where η is the normalizer variable. $x_{t}$ is a Boolean random variable that represents whether the landmark is present at time *t*. Moreover, $z_{1:t}$ is a sequence of Boolean random variables that indicate whether the landmark is detected up to time *t*. In this paper, we use the assignment results of the landmark as the detection outputs. In other words, if the landmark is assigned with observations, it is assumed that the landmark is detected. Otherwise, we assume that the landmark is not detected.   
**Algorithm 2:** State estimation procedure for an existent landmark on the HD map.
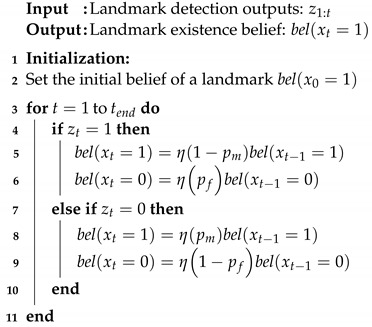


As described in line 3–11 of Algorithm 2, $\mathrm{bel}\left(x_{t}\right)$ is recursively updated in accordance with $z_{t}$ until the end of time $t_{end}$.
(4)$$\begin{array}{c} p_{m}=p\left(z_{t}=0|x_{t}=1\right)\\ p_{f}=p\left(z_{t}=1|x_{t}=0\right) \end{array}$$
In this process, two parameters $p_{m}$ and $p_{f}$ are utilized. $p_{m}$ and $p_{f}$ correspond to the missed detection rate and false alarm rate in Equation (Equation 4), respectively. Therefore, if $\mathrm{bel}\left(x_{t_{end}}=1\right)$ is larger than the initial belief $\mathrm{bel}\left(x_{0}=1\right)$, this landmark is estimated as *Normal*. Otherwise, it is estimated to be *Deleted*.

Finally, the proposed state estimation is performed several times because our crowdsourced data is divided by the vehicle index, as shown in Section 3.2. In every estimation, the state of the landmark is decided independently, and all estimation results are merged as in Equation (Equation 5).
(5)$$\frac{N_{n}}{N_{n}+N_{d}}>thr_{normal}$$

$N_{n}$ and $N_{d}$ represent the numbers of *Normal* and *Deleted* estimations, respectively. When the number of total estimations is not sufficient, the classification of the landmark is skipped. If the fraction between $N_{n}$ and the total number of estimations is greater than the threshold $thr_{normal}$, the landmark is classified as *Normal*. In the opposite case, the landmark is classified as *Deleted*. The reliability and robustness of the classification are further improved by integrating multiple estimations. All HD map landmarks are classified using the proposed method.

*New* and *Outlier* are classified using the clustered candidates from Section 3.4.2. Suppose that the number of traversals from the crowdsourced data is *N*. For each candidate, *New* and *Outlier* are determined by comparing *N* with the number of landmarks in each candidate $n_{c,i}$, as shown in Equation (Equation 6).
(6)$$\frac{n_{c,i}}{N}\cdot \left(1-\frac{1}{\sqrt{N}}\right)>thr_{new}$$i: candidate index


Sometimes, $\frac{n_{c,i}}{N}$ is extremely large because of the small value of *N*. In this case, the candidate that is not fully clustered may be classified as *New*. Thus, to prevent false classification, $1-\frac{1}{\sqrt{N}}$ is multiplied as shown in Equation (Equation 6). In summary, if the ratio of $n_{c,i}$ to *N* multiplied by $1-\frac{1}{\sqrt{N}}$ is larger than the threshold $thr_{new}$, then the $i_{th}$ candidate is classified as *New*. Otherwise, the $i_{th}$ candidate is classified as *Outlier*.

#### 3.5.2. Landmark State Update

The landmark update step reflects the classification results from Section 3.5.1 on the HD map. In this step, the landmarks are updated with reference to each sub-map as described in Section 3.2. *Normal* is maintained on the map while *Deleted* is removed. Furthermore, *New* is newly added on the updated map by means of Bézier curve algorithm [49], a widely used lane fitting method.

If any changes take place on each sub-map, this map will be automatically reported. Since the map updates should be carefully carried out, a human expert responsible for map management makes the final decision. After investigating the new HD map candidates and recent road images, valid updates are manually selected.

## 4. Experimental Result

To validate our proposed method, we utilized ROD collected from five test areas in urban environments. The amount of data is shown in Figure 2. We present our results in three ways. First, we determined the amount of crowdsourced data that could be used throughout the observation correction step. Second, the performance of our method was evaluated by comparing it with our previous method [44]. Finally, we demonstrated the number of sub-HD maps that could be reliably updated using the proposed method. In this study, we referred to street views from the Kakao map [50] and Naver map [51] to manually label the ground truth.

### 4.1. Validation Check Results

Table 2 shows that pose correction can increase the amount of utilized crowdsourced data after the validation check step. Although slight differences were observed between the test areas, the number of available keyframes increased by 5–8%. The number of surviving lane observations also increased by 6–11%. Given the considerable amount of crowdsourced data, the average increase of 8.7% in lane observations indicates that the effect of the pose correction step is useful. Therefore, this result implies that crowdsourced data can become more robust to environmental disturbances through the pose correction step.

### 4.2. Qualitative Evaluation

Figure 6 shows a qualitative comparison of map updates for the previous and proposed methods. Figure 6a represents the aerial image of the road environment from Test area 1 as the ground truth of environmental changes. Figure 6b presents the HD map before update. In Figure 6c,d, map updates by the previous and proposed methods are shown, respectively. In addition, *Normal* is depicted by a white dotted line and *Deleted* is indicated by a yellow dashed line. *New* corresponds to a cyan dash-dotted line.

In Figure 6c, the previous method resulted in some errors. An overlap existed between the double *New* lanes, which indicates that one actual *Outlier* is incorrectly predicted as *New*. Moreover, an actual *Deleted* lane was not found, as represented by the dotted ellipse. Several *Normal* lanes were falsely predicted as *Deleted*, as shown in the dotted squares.

However, Figure 6d shows that the proposed method updated the *New* lane with comparable quality to the ground truth in Figure 6a. Furthermore, all the *Normal* and *Deleted* landmarks were correctly detected. These findings imply that the proposed method outperforms the previous method in the detailed detection of road changes.

### 4.3. Quantitative Evaluation

Table 3 and Table 4 present a quantitative comparison between the previous and proposed methods for each of the five test areas. The quantitative measures for the *Normal-Deleted* and *New-Outlier* classifications are listed in Table 3 and Table 4, respectively. The measures include precision, recall, f1-score, and specificity.

From Table 3, it can be observed that the precision values for most areas were close to 1. This was due to a biased ratio between the landmark classes in the test areas. Indeed, the number of unchanged lanes was usually much larger than that of removed lanes in road environments. Therefore, we focused on other measures, such as specificity and recall, to evaluate the quality of the updated maps.

Both the previous and proposed methods exhibited similar specificity values within 7%. This means that there is no significant difference between the two methods in detecting *Deleted* landmarks. In the case of Test area 3, the specificity was not computed because of the absence of removed lanes. In contrast, the proposed method outperformed the previous method in terms of the recall index. Specifically, the recall values increased by 12–20%. This implies that the proposed method can extract *Normal* lanes more accurately without inappropriate deletions.

It is clear that the accurate detection of *Deleted* lanes is essential for HD map updates. However, from the viewpoint of autonomous driving, the prevention of the improper removal of *Normal* lanes could be more important. In urban environments, the missed detection of *Deleted* landmarks could be a simple problem because occlusion by pedestrians and vehicles occurs frequently. However, incorrect deletion of *Normal* lanes can decrease localization performance owing to repeated observations of unexpected landmarks.

As described in Table 3, the proposed method showed similar specificity values when compared to the previous method. However, there were significant improvements in the recall values. Therefore, it can be concluded that the proposed method shows superior performance over the previous method in *Normal-Deleted* classification.

In Table 4, we can see that the proposed method outperformed the previous method in terms of the overall measures. In particular, the specificity values of the proposed method were close to 1. This means that most of the *Outlier* lanes can be filtered out from the clustered candidates.

Although the recall values increased by 4–25%, the precision values increased by 30–57%. This marked increase in precision implies that the proposed method can detect *New* landmarks more precisely. Due to the low quality of crowdsourced data, the exact detection of *New* landmarks could be essential. When updating *New* landmarks, many inaccurate lanes can be marked. Considering the frequent updates of HD maps, focusing on updating precise landmarks can be helpful. Thus, we can also confirm that the proposed method performs better in *New-Outlier* classification.

### 4.4. Updated HD Maps

Figure 7 shows reliable, updated maps from our proposed method. Figure 7a shows the ground truths of the road environments using the corresponding aerial images. Since many aerial images are outdated, deleted and newly added lanes are represented by yellow dashed lines and cyan dash-dotted lines, respectively. Furthermore, Figure 7b,c represent the previous and updated HD maps, respectively. In Figure 7b,c, the landmark descriptions are identical to those in Figure 6.

In total, our method suggested 45 updated sub-maps from the five test areas. We automatically detected all points with the possibility of even a slight change from a large amount of data and made a suggestion to a manager. The manager finally checked recent images of the proposed point to approve the map update. How aggressive or conservative map updates should be is a strategic issue that must be determined by the target application of the map. Therefore, it was designed to present as many candidates as possible because it is safe to ensure that managers check for all points that might have changed.

Finally, these updated maps were considered reliable only when the changed environments were clearly represented. As shown in Figure 7c, the number of valid updates was 14. Although a simple human intervention was required to investigate recent road images, the proposed automated procedure for map updates remarkably reduced the overall cost of HD map updates. Therefore, frequent and timely map updates were made possible using the proposed method.

## 5. Conclusions

In this study, we proposed the useful HD map update strategy using crowdsourced data. The aim of our research is a frequent and accurate update of HD maps without manual work by human experts. Well-organized computational procedures are established to deal with uncertain crowdsourced data. Especially two main algorithms are designed to update traffic lanes. First, convex hull-based landmark searching is leveraged to prevent uncertain observation assignments. As a next step, recursive Bayesian estimation is used to track the possibility of changes in a robust manner.

To conclude, the proposed map update strategy was tested successfully using a large amount of crowdsourced data collected by many city buses and taxis in Seoul. The total travel distance and driving time of all the crowdsourcing vehicles are approximately 89,620 km and 3600 h every day. Therefore, we can collect plenty of observations for broad urban environment. Our results show that we can suggest an accurate map update with a probability of about 31% by leveraging crowdsourced data obtained over a single day. Since the map updating procedure can be automatically carried out with a simple human intervention, frequent updates are available with significantly reduced costs.

As a further step for our research, we hope to focus on how much the amount of crowdsourced data is adequate for timely map updates. Because an enormous amount of data can be collected, the reference for minimum data usage would prevent unnecessary waste of time and resources in map maintenance. In addition, we plan to extend our map update strategy to other types of landmarks, such as traffic signs and traffic lights, which also plays an important role in localization for autonomous driving.

## Figures and Tables

**Figure 1 sensors-23-00438-f001:**
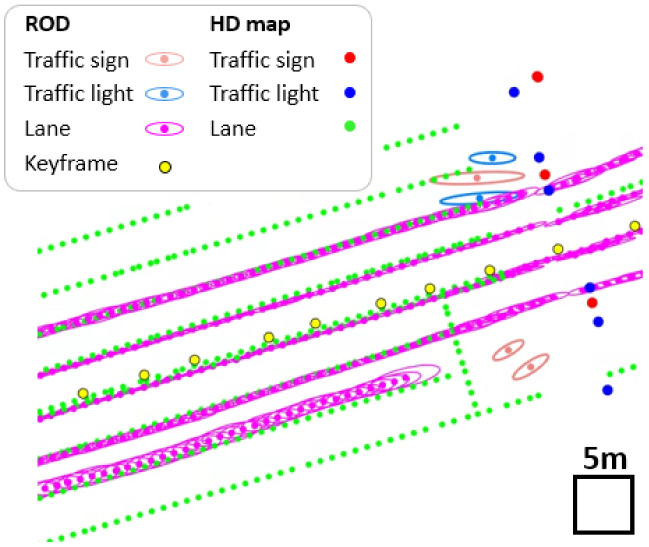
ROD visualization of a single drive.

**Figure 2 sensors-23-00438-f002:**
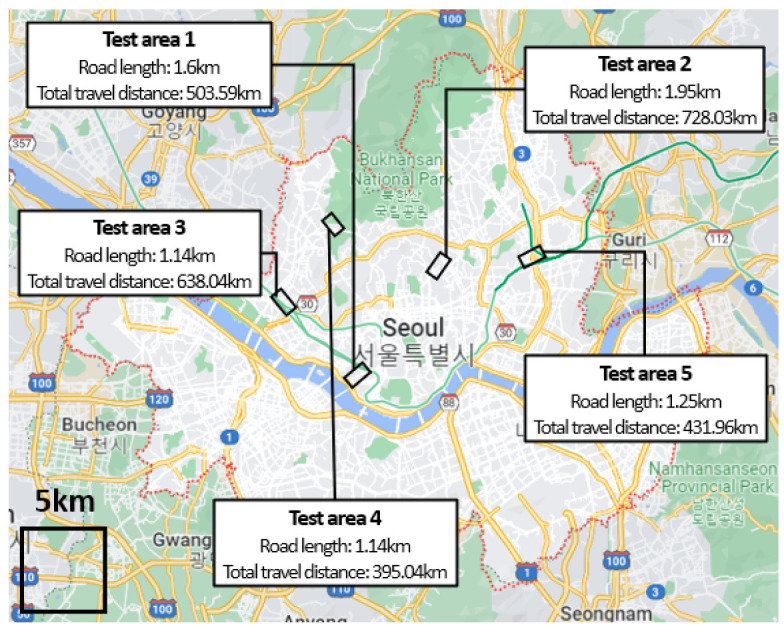
Quantitative information of the ROD collected from randomly sampled five test areas. Road length refers to the length of the road used for data collection in each test area. The total travel distance is the sum of the travel distances by multiple vehicles.

**Figure 3 sensors-23-00438-f003:**
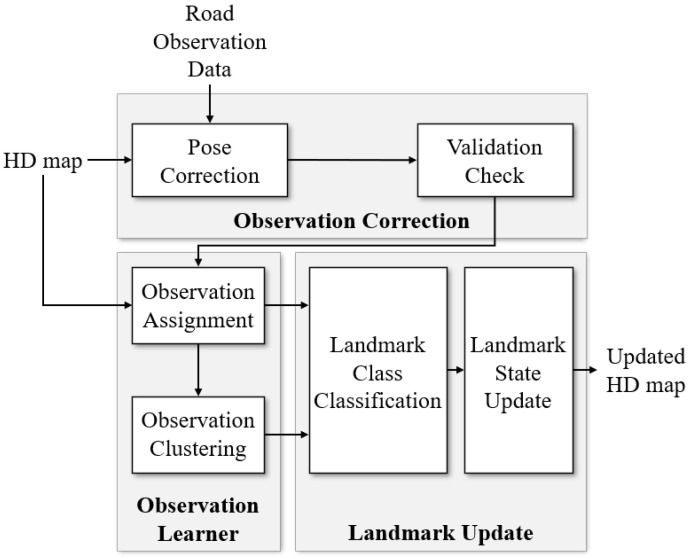
Overview of the HD map update scheme.

**Figure 4 sensors-23-00438-f004:**
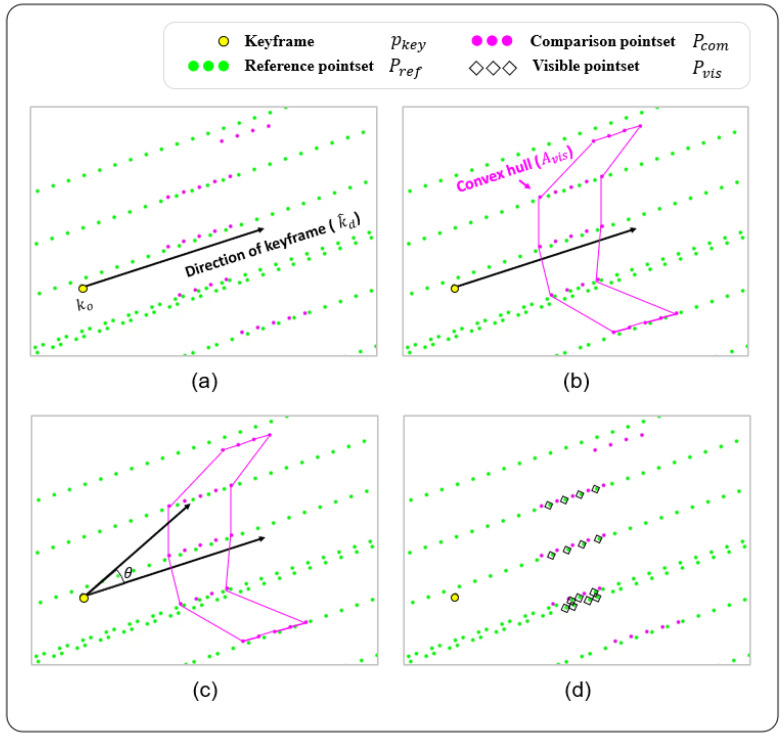
Illustration of searching procedure for visible pointset: If a reference landmark point from $P_{ref}$ lies inside the convex hull of $P_{com}$ and exists within a certain angle range referenced to a keyframe, this reference point is considered visible. (**a**) HD map and crowdsourced data. (**b**) Convex hull construction. (**c**) Visible point selection. (**d**) Visible pointset.

**Figure 5 sensors-23-00438-f005:**
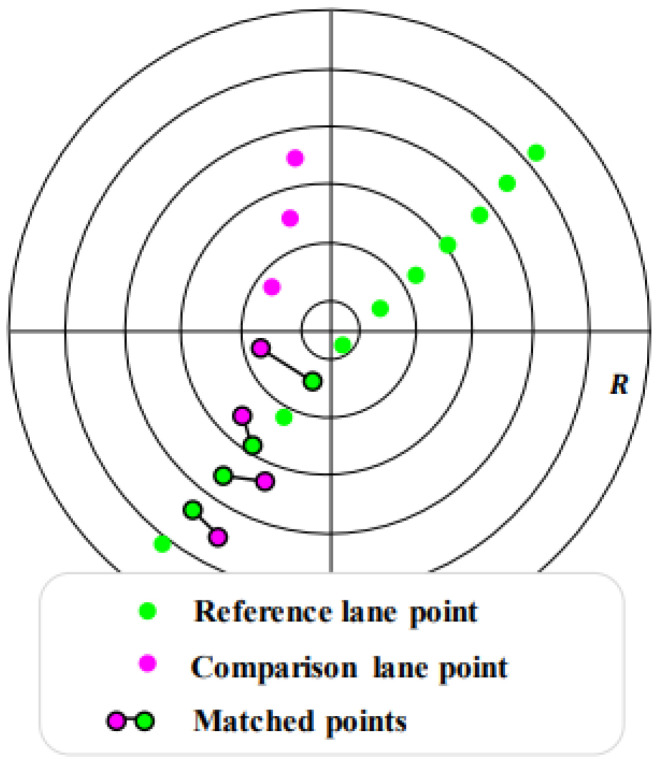
Illustration of shell structure with continuous landmarks. The shell structure is partitioned by sectors surrounded by black lines. When the reference and comparison lane points exist together in each sector, they are matched in a way that minimizes the distance between two points.

**Figure 6 sensors-23-00438-f006:**
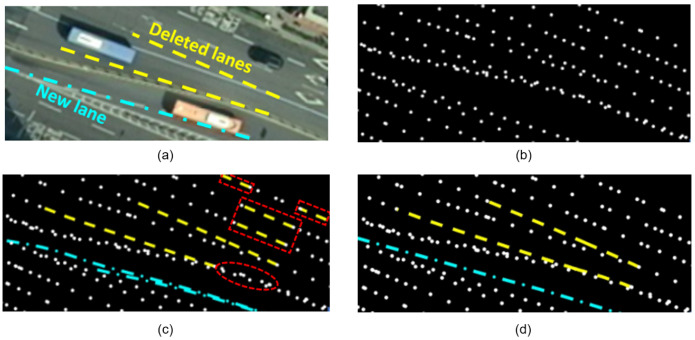
HD map update results of the previous and proposed method for Test area 1. (**a**) Aerial image of a road environment. (**b**) HD map before update. (**c**) HD map updated by the previous method. (**d**) HD map updated by the proposed method. An aerial image is presented as the ground truth in (**a**). In (**b**–**d**), different landmark classes are represented using different line styles and colors. *Normal*, *Deleted*, and *New* correspond to a white dotted line, yellow dashed line, and cyan dash-dotted line, respectively. In (**c**), dotted squares represent the areas where actual *Normal* lanes were falsely predicted as *Deleted*. In addition, a dotted ellipse shows the area where an actual *Deleted* lane was predicted as *Normal*.

**Figure 7 sensors-23-00438-f007:**
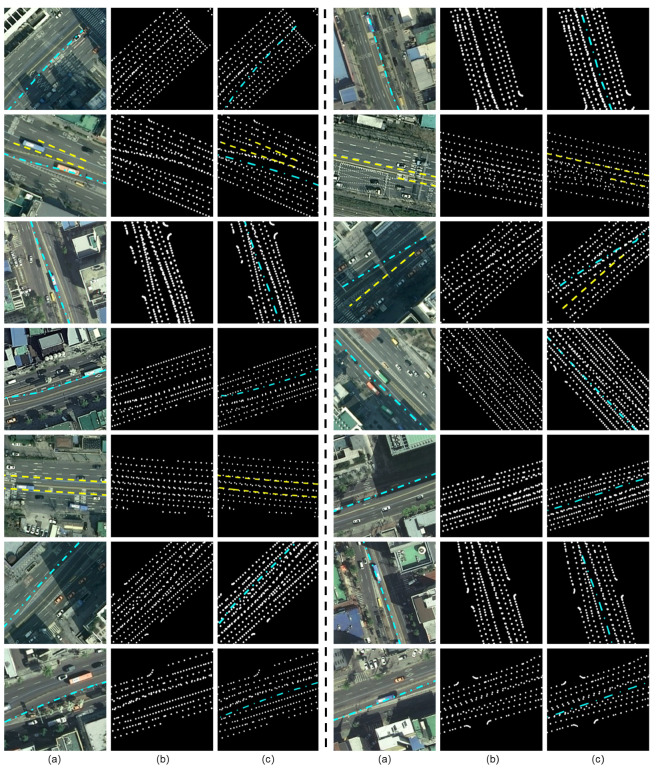
Updated HD maps from five test areas. (**a**) Aerial image. (**b**) HD map before updating. (**c**) Updated HD map. The algorithm automatically detected the changed environments and suggested map updates at 45 locations. Finally, a human manager approved 14 updates after manually investigating road images. The presented aerial images may not accurately match HD maps, because many images are outdated. Thus, deleted and newly added lanes are indicated by the yellow dashed and cyan dash-dotted lines, respectively. All HD maps represent different landmark classes with different line styles. *Normal*, *Deleted* and *New* correspond to white dotted, yellow dashed, and cyan dash-dotted lines, respectively.

**Table 1 sensors-23-00438-t001:** Definition of landmark classes.

	Landmark Existence	
**Class**	**Before Update**	**After Update**
*Normal*	existent	existent
*Deleted*	existent	non-existent
*New*	non-existent	existent
*Outlier*	non-existent	non-existent

**Table 2 sensors-23-00438-t002:** The ratio of utilized crowdsourced data after the validation check.

	Keyframe		Lane Observation	
	**Without** **Pose Corrention**	**With** **Pose Correction**	**Without** **Pose Correction**	**With** **Pose Correction**
Test area 1	50.38%	55.30%	63.83%	70.86%
Test area 2	43.18%	50.42%	51.91%	61.60%
Test area 3	40.22%	47.59%	45.63%	55.50%
Test area 4	50.55%	58.69%	64.84%	75.89%
Test area 5	32.46%	37.02%	38.74%	44.79%

**Table 3 sensors-23-00438-t003:** Quantitative evaluation results of the *Normal-Deleted* classification for the previous and proposed methods.

	*Normal-Deleted* Classification							
	**Precision**		**Recall**		**F1-Score**		**Specificity**	
	**Previous** **method**	**Proposed** **Method**	**Previous** **Method**	**Proposed** **Method**	**Previous** **Method**	**Proposed** **Method**	**Previous** **Method**	**Proposed** **Method**
Test area 1	0.99	0.99	0.72	**0.89**	0.83	0.94	**0.82**	0.75
Test area 2	0.95	0.95	0.72	**0.84**	0.82	0.89	**0.60**	0.54
Test area 3	1.00	1.00	0.68	**0.88**	0.81	0.94	-	**-**
Test area 4	0.98	0.99	0.69	**0.85**	0.81	0.91	0.63	**0.67**
Test area 5	0.99	0.99	0.70	**0.88**	0.82	0.94	**0.5**	**0.5**

**Table 4 sensors-23-00438-t004:** Quantitative evaluation results for the *New-Outlier* classification for the previous and proposed methods.

	*New-Outlier* Classification							
	**Precision**		**Recall**		**F1-Score**		**Specificity**	
	**Previous** **Method**	**Proposed** **Method**	**Previous** **Method**	**Proposed** **Method**	**Previous** **Method**	**Proposed** **Method**	**Previous** **Method**	**Proposed** **Method**
Test area 1	0.29	**0.86**	0.44	0.60	0.35	0.71	0.91	**0.95**
Test area 2	0.33	**0.63**	0.38	0.63	0.35	0.63	0.93	**0.94**
Test area 3	0.38	**0.75**	0.55	0.67	0.44	0.71	0.92	**0.97**
Test area 4	0.26	**0.63**	0.38	0.50	0.31	0.56	0.84	**0.94**
Test area 5	0.26	**0.75**	0.56	0.60	0.36	0.67	0.84	**0.98**

## Data Availability

Not applicable.

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
