# Peer review of "Frequent and Automatic Update of Lane-Level HD Maps with a Large Amount of Crowdsourced Data Acquired from Buses and Taxis in Seoul"

_sensors, 2022, doi:10.3390/s23010438_

Round 1
Reviewer 1 Report
The article needs substantial revision, as it appears lacking in some aspects.
1. It is necessary to clarify the paper's objectives and the type of high-density maps which need to be updated. The title and abstract are too generic concerning the maps to be treated. These maps in the text appear related to the roadway elements and, above all, to the lanes' layout.
2. The authors need to frame the available literature better. The proposed literature review does not seem too thorough, and it is not clear what gaps exist that lead to the development of the proposed procedure
3 The authors need to clarify what is better about this procedure than other ways that could be used for the updating of the maps (it would be helpful to highlight some comparisons).
4 For which application purpose is the updating of the maps proposed? The authors firstly exclude that it can be used for autonomous vehicles, which need real-time systems, but after, they reconsider the opportunity of using it for this type of vehicle. They need to detail better their intentions concerning these issues.
5 The method is explained qualitatively with some quantitative arguments (some metalanguage algorithms and mathematical relations). A more straightforward explanation for the reader would be obtained with the possibility of following what has been described step by step with examples.
6 The conclusions are scarce. The authors need to better argue about the research objectives, the materials and methodologies used, the results obtained, their usability in applications, the proposed procedure's limits, and future developments, highlighting the value of their work.
Author Response
The authors would like to express their sincere gratitude to the reviewers for their considerate and valuable comments on this paper. The manuscript is substantially revised according to the reviewer’s comments. Modified sentences are highlighted by using yellow-shaded fonts in the manuscript. Followings are the answers and the list of modifications for the questions and comments. The reviewer’s comments are labeled according to the order of appearance.

Reviewer 2 Report
1.High-definition maps are an important factor in the field of autonomous driving vehicles. The automatic update method of high-definition maps proposed in this paper can solve the potential danger to automatic driving when some road sections are updated. Overall, the method proposed in this paper is feasible, the amount of collected data is large, and the designed method can effectively process the data.
2.What is the regularity of the influence of the number of participating vehicles and the number of travel mileage on the update results? Of course, the more the better, but there needs to be a certain balance between the timeliness of actual use requirements and the quality of updates.
Author Response

(The authors gave the same response as above.)

Reviewer 3 Report
In this submission, the authors propose a new strategy to update HD maps from large amounts of low-quality crowdsourced sensor data. The work is well motivated, and overall presentation is of good quality. However, there are a few minor issues:
1. "Only the changed environments are successfully extracted". Both in the abstract and line 86 of the introduction. The use of "only" in this context could imply that something else was not successfully extracted, which does not seem to be the intention.
2. The abbreviations GNSS and RTK-GNSS are never defined, whereas most other abbreviations are.
3. In Figure 4 and the description of Algorithm 1, the authors talk about the convex hull of the comparison pointset. However, the outlined area in Figure 4 does not correspond to the convex hull of the comparison pointset.
4. Also in Figure 4, the visible pointset is denoted Pviw, which should be Pvis according to the text description.
5. Line 321+322 "the exact detection of new landmarks is more important than their entire detection". It is not entirely clear to me what is meant with this statement, is the exact position of the new landmark more important than detecting the landmark at all? If so, why?
6. In Section 4.4 it is noted that out of 45 suggested updates, only 14 were accepted. It would be good to include the reason for the rejections. Were the other suggested updates wrong?
7. The conclusion is relatively terse. The authors may want to include remaining challenges and opportunities for future research.
Author Response

(The authors gave the same response as above.)

Round 2
Reviewer 1 Report
All my observations have been taken into account in the new version of the paper, which is improved in the presentation and exposure of the research.
In my opinion, the paper is suitable for publication in its current version.